# Structure and Function of Soil Bacterial Communities in the Different Wetland Types of the Liaohe Estuary Wetland

**DOI:** 10.3390/microorganisms12102075

**Published:** 2024-10-16

**Authors:** Yunlong Zheng, Fangli Su, Haifu Li, Fei Song, Chao Wei, Panpan Cui

**Affiliations:** 1College of Forestry, Shenyang Agricultural University, Shenyang 110866, China; 2College of Water Conservancy, Shenyang Agricultural University, Shenyang 110866, China; 3Liaoning Panjin Wetland Ecosystem National Observation and Research Station, Shenyang 110866, China; 4Liaoning Shuangtai Estuary Wetland Ecosystem Research Station, Panjin 124112, China; 5Liaoning Provincial Key Laboratory of Soil Erosion and Ecological Restoration, Shenyang 110866, China

**Keywords:** estuary wetland, bacterial community, bacterial diversity, functional genes

## Abstract

Soil bacterial communities play a crucial role in the functioning of estuarine wetlands. Investigating the structure and function of these communities across various wetland types, along with the key factors influencing them, is essential for understanding the relationship between bacteria and wetland ecosystems. The Liaohe Estuary Wetland formed this study’s research area, and soil samples from four distinct wetland types were utilized: suaeda wetlands, reed wetlands, pond returning wetlands, and tidal flat wetlands. The structure and function of the soil bacterial communities were examined using Illumina MiSeq high-throughput sequencing technology in conjunction with the PICRUSt analysis method. The results indicate that different wetland types significantly affect the physical and chemical properties of soil, as well as the structure and function of bacterial communities. The abundance and diversity of soil bacterial communities were highest in the suaeda wetland and lowest in the tidal flat wetland. The dominant bacterial phyla identified were Proteobacteria and Bacteroidota. Furthermore, the dominant bacterial genera identified included *RSA9*, *SZUA_442*, and *SP4260*. The primary functional pathways associated with the bacterial communities involved the biosynthesis of valine, leucine, and isoleucine, as well as lipoic acid metabolism, which are crucial for the carbon and nitrogen cycles. This study enhances our understanding of the mutual feedback between river estuary wetland ecosystems and environmental changes, providing a theoretical foundation for the protection and management of wetlands.

## 1. Introduction

Estuarine wetlands are the transitional zones between rivers and the ocean and serve as critical links between terrestrial and marine ecosystems [1]. These wetlands are characterized by rich biodiversity, high biological productivity, and a wide range of ecosystem service functions; thus, they are vital ecological environments within larger ecosystems [2]. Soil microbial communities, as integral components of soil ecosystems, remarkably influence the structure and function of wetland ecosystems [1,3,4]. In this complex ecosystem, soil microbes decompose organic matter and participate in the biogeochemical cycling of elements, thereby playing a crucial role [5,6,7,8]. Soil microbial communities encompass a diverse array of bacterial groups that are essential for ecological processes, including the decomposition of organic matter, the cycling of elements such as carbon, nitrogen, and phosphorus, the emission of greenhouse gases, and the transformation of nutrients. Consequently, bacteria can serve as indicators of the stability of microbial communities and the soil quality, and can reflect the sensitivity of soil environments [8]. Therefore, a comprehensive understanding of bacterial communities and their ecological functions in wetland soils is crucial for the protection and restoration of estuarine wetlands.

The structure and function of soil bacterial communities in estuarine wetlands are influenced by various environmental factors and are highly sensitive and responsive to external changes [9,10,11]. Substantial advancements have been made in recent years in the study of wetland soil microbiota which illuminate the structure and composition of microbial communities across different wetland soils and the factors that influence them. Variations in wetland types can lead to differences in the physical and chemical properties of the soil, which subsequently affect the structure and function of soil bacterial communities, and vice versa [8]. Soil microbial communities considerably influence plant metabolism and their resistance to pathogens. Factors such as soil salinity, moisture, and temperature play crucial roles in determining the composition and diversity of bacterial communities. Additionally, vegetation types and soil moisture conditions are essential in shaping the structure of these microbial communities [12,13]. Research conducted by [14] on the structure and function of bacterial communities across various wetland types has shown that the Proteobacteria phylum is a prevalent taxon in alpine, inland, and coastal wetlands. However, remarkable differences are observed in the structures of bacterial communities across diverse types of wetlands. A study conducted in the freshwater wetlands of the Yellow River Delta [15] indicated that the abundance of sulfate-reducing bacteria, marmoricola, and nitrifying bacteria decreases with an increase in spatial distance from the riverbank. In addition, complex environmental factors, such as vegetation [16], pH [17], and nutrient levels [18,19], may directly shape the bacterial communities inhabiting the wetland soil and alter their composition. Wetlands host diverse multifunctional microbial communities and may contribute to ecological balance, playing a key role in driving biogeochemical cycles. In the carbon cycle, for instance, methane is produced by methanogenic microorganisms and serves as a key substrate that is oxidized by methanotrophs under anaerobic or aerobic conditions [20]. The nitrogen cycle consists of multiple processes, namely, nitrification, denitrification, nitrogen fixation, ammonification, and nitrate reduction, which rely on the synergistic action of various bacterial species and functional genes [3]. Previous studies have mostly focused on a single factor or a specific type of wetland, such as reed wetlands [21], suaeda wetlands [16], and intertidal zones [22]. However, in the unique environment of estuarine wetlands, the response of soil bacterial communities to changes in environmental factors and wetland types, and how these changes affect the diversity of the bacterial community structure and function, is insufficiently understood. Comprehensive research is urgently needed to elucidate the integrated effects of various environmental factors and wetland types on the diversity and function of soil bacterial communities. Our research centered on the Liaohe Estuary Wetland, with objectives encompassing four distinct wetland types within this region. We analyzed the differences in bacterial community structure and function by comparing soil samples from these various wetland types. We aimed to enhance our knowledge of the soil characteristics of the different wetland types in the same area, thereby contributing to the study of microorganisms. Based on the established relationships between soil physicochemical properties and microbial communities, we hypothesized that the structure and function of soil bacterial communities would significantly differ between various wetland types, including reed wetlands, suaeda wetlands, pond returning wetlands, and tidal flat wetlands. Furthermore, we anticipated that these differences would be closely associated with key processes, such as soil salinity, pH levels, organic matter content, and nutrient cycling. To test these hypotheses, we elucidated the structure, diversity, and function of the soil bacterial communities across the four typical wetland types in the Liaohe Estuary Wetland. This research utilized Illumina MiSeq sequencing to analyze the composition and structure of the bacterial communities and PICRUSt to predict their metabolic functional profiles and to investigate the mechanisms by which varying wetland environments influence soil microbial communities and their functions. This study offers new insights into the structure and function of bacterial communities in estuarine wetlands in relation to environmental changes.

## 2. Materials and Methods

### 2.1. Research Scope and Sampling

The study area was located in the Liaohe Estuary Wetland (40°45′~41°06′ N, 121°28′~121°59′ E), situated in the Liaohe Delta of Northeast China. This region has a temperate semi-humid monsoon climate, with an average annual temperature of 8.5 °C and an average annual precipitation of 650 mm [3]. The natural wetland types include tidal flat wetlands, suaeda wetlands, reed wetlands, and many aquaculture pond returning wetlands. The main plant communities are reeds and suaeda [13,23].

The field investigation and sampling were conducted in September 2023. During this period, the average temperature in the study area was 23 °C, with 46.2 mm of precipitation. The daylight duration was recorded at 141.8 h, and the humidity levels ranged from 48% to 85%. Twelve plots were established which encompassed the four types of wetlands: suaeda wetlands, reed wetlands, pond returning wetlands, and tidal flat wetlands. Each wetland type had three plots. Soil samples were collected from each plot using a five-point sampling method with a depth of 0–10 cm. The soil from the five sampling points in each plot was mixed on site to form a composite soil sample, named with a combination of letters and numbers. After removing plant debris and stones, each sediment sample was divided into several parts. A portion of the soil was air dried and further analyzed for its physicochemical properties after being sieved through a 100-mesh screen. Another part was stored at −80 °C in a freezer for subsequent DNA extraction and sequencing.

### 2.2. Determination of Soil Physical and Chemical Properties

The gravimetric method was used to determine each sediment’s water content, and a drying oven (Thermo Fisher, OMH750, Thermo Fisher Scientific, Waltham, MA, USA) and electronic balance (DELIXI, DLX-A8, Delixi Electric Ltd., Leqing, China) were used. The pH and salinity were analyzed using a multiparameter analyzer (INESA, DZS-708, Shanghai Precision Scientific Instrument Co., Ltd., Shanghai, China) with the sediment solution, in which the ratio of sediment to deionized water was 1:2.5. Total carbon (TC) and total nitrogen (TN) were detected using a TOC analyzer (Elemen-tar, enviro TOC, Elementar Analysensysteme GmbH, Hanau, Germany) [24]. Total phosphorous (TP) was analyzed via the alkali fusion-Mo-Sb anti-spectrophotometric method [25]. Total potassium (TK) was analyzed using a flame atomic absorption spectrophotometer (TAS-990 SUPER AFG, PG Scientific, Inc., Beijing, China) [26].

### 2.3. DNA Extraction, PCR Amplification, and Illumina MiSeq Sequencing

Genomic DNA was extracted from each soil sample (samples) using the E.Z.N.A.™Mag-Bind Soil DNA Kit (Omega Bio-tech, Norcross, GA, USA). High-throughput sequencing targeted the V3–V4 region of the 16S rRNA gene, which was amplified using the barcoded primers 341F (5′-CCTACGGGNGGCWGCAG-3′) and 805R (5′-GACTACHVGGGTATCTAATCC-3′), following the PCR amplification protocol described in [27]. The amplicons were purified and quantified using the Qubit 2.0 DNA ASSay Kit (Thermo Fisher Scientific, Waltham, MA, USA) [28]. Subsequently, the purified amplicons (20 pmol) from each sample were sequenced on the Illumina MiSeq platform PE300 at Sangon Biotech (Shanghai, China).

### 2.4. Sequence Data Processing

All raw sequences from the samples were processed using the DADA2 plugin within Qiime2 (2022.2) software for quality control purposes to filter out low-quality sequences. Denoising was performed to correct sequencing errors, and the sequences were merged (this step was skipped for single-end data). Chimera detection and removal were conducted to ensure the sequences were non-chimeric, resulting in the formation of OTUs. Representative sequences for each OTU were selected and matched against the Greengenes database, version 13_8, to obtain the taxonomic annotation information. The number of sequences at the phylum level was determined for each sample based on the absolute abundance of OTUs and the annotation information. One-way ANOVA was used to assess the differences in the physicochemical properties of the soil samples from the different sites. A *p* < 0.05 indicated statistically significant differences. Pearson correlation tests were conducted using SPSS (22.0) to calculate the correlations between soil properties. The α-diversity of the soil bacterial communities was analyzed using Qiime2, including metrics such as observed OTUs (observed_features), Shannon, and Faith’s phylogenetic diversity. The Wilcoxon test was employed to evaluate the significance of the α-diversity differences between the sampling sites. The beta diversity was assessed using R (4.3.3), including the non-metric multidimensional scaling (NMDS) method. Environmental analyses and Spearman correlation evaluations were performed with R software. Metabolic functions were predicted based on the 16S rRNA gene sequencing data using PICRUSt analysis (https://picrust.github.io/picrust/, accessed on 6 December 2023). The predicted genes were annotated according to the KEGG Orthology database.

## 3. Results

### 3.1. Soil Physicochemical Properties in Different Wetland Types

The soil physicochemical properties exhibited significant differences between the four wetland types in the study area (*p* < 0.05; Figure 1). The soils were predominantly alkaline (pH > 8.0), and the highest alkalinity was observed in the suaeda wetlands. Notable variations in the soil salinity were present, and the highest levels were recorded in the pond returning wetlands. Conversely, the reed wetlands demonstrated the highest total nitrogen (TN) content alongside the lowest total phosphorus (TP) content. Additionally, the total carbon (TC) content was significantly lower in the pond returning wetlands than in the three other wetland types.

### 3.2. Bacterial Community Richness and Diversity Indices of Soil in Different Wetland Types

Low-quality sequences were automatically filtered out using the MiSeq high-throughput sequencing platform, resulting in 2,092,024 trimmed sequences detected from 12 soil samples across the four types of wetlands. Each wetland type yielded between 130,878 and 192,837 sequences, with an average base length of 423 bp. At the phylum level, 29,593 operational taxonomic units (OTUs) were identified, and their distribution across the four wetland types in descending order was as follows: suaeda wetland > reed wetland > pond returning wetland > tidal flat wetland (Figure 2a). A Venn diagram comparison of the bacterial OTUs in the soil of the four wetland types revealed 143 shared OTUs among all four types. The numbers of unique OTUs in the suaeda wetland, reed wetland, pond returning wetland, and tidal flat wetland were 9379, 9763, 9763, and 4852, respectively. The pond returning and tidal flat wetlands exhibited the highest number of shared OTUs, totaling 1251; meanwhile, the reed and pond returning wetlands had the fewest shared OTUs, amounting to 375 (Figure 2b).

The soil species richness and diversity significantly varied between the different wetland types (Figure 2a). The reed wetland demonstrated the highest Shannon index of 10.39. The suaeda wetland showed the highest Chao1 index of 4625.61, which was significantly greater than the values observed in the tidal flat and pond returning wetlands. The NMDS analysis of the soil microbial communities across the four wetland types is presented in Figure 2c. The stress value for the suaeda wetland was 0.0721, which is below the threshold of 0.1, indicating that the analysis possessed satisfactory explanatory significance. Furthermore, the NMDS analysis revealed a clear distance and significant differences between the soil microbial communities of the four wetland types. The tidal flat and reed wetlands exhibited a substantial gap in the NMDS1 direction, whereas the pond returning and suaeda wetlands displayed a distinct gap in the NMDS2 direction.

Redundancy analysis (RDA) was employed to examine the relationship between the soil bacterial community structure and various physicochemical factors. The lengths of the arrows in the RDA plot indicate the influence of these physicochemical factors on the ordination. Figure 2d shows that water content (WC), soil extractable carbon (SEC), salt, total phosphorus (TP), and total carbon (TC) were significant variables, as confirmed with permutation tests, which yielded a significance level of *p* < 0.05.

### 3.3. Distribution and Composition of Soil Bacterial Communities in Different Wetland Types

The different wetland types exhibited variations in the relative abundances of bacterial phyla and genera (Figure 3). Across all samples, 94 bacterial phyla and 1 archaeal phylum were identified, and 18 phyla demonstrated an average abundance greater than 1%. At the phylum level (Figure 3b), the ten dominant phyla in descending order of abundance were as follows: Proteobacteria (26.43% to 33.92%), Bacteroidota (9.39% to 18.25%), Gemmatimonadota (8.20% to 19.81%), Actinobacteriota (2.17% to 24.41%), Acidobacteriota (1.74% to 9.41%), Halobacteriota (0.03% to 15.93%), Myxococcota_A_473307 (2.23% to 5.52%), Chloroflexota (0.94% to 6.03%), Desulfobacterota_I (0.75% to 5.56%), and Desulfobacterota_G_459546 (0.99% to 4.03%). Proteobacteria and Bacteroidota were the most dominant phyla, followed by Gemmatimonadota and Actinobacteriota. The relative abundance of Proteobacteria was highest in the suaeda wetland, followed by the pond returning wetland, tidal flat wetland, and reed wetland. The relative abundance of Bacteroidota was significantly lower in the reed wetland than in the pond returning wetland, tidal flat wetland, and suaeda wetland. Meanwhile, the relative abundance of Actinobacteriota was significantly higher in the pond returning wetland, tidal flat wetland, and suaeda wetland. Furthermore, the relative abundance of Gemmatimonadota was markedly higher in the tidal flat wetland compared with the pond returning wetland, reed wetland, and Suaeda wetland. The relative abundance of Halobacteriota considerably varied between the four wetland types. The highest levels were observed in the pond returning wetland. Myxococcota_A_473307, Thermoproteota (1.14% to 3.45%), Verrucomicrobiota, and Desulfobacterota_B exhibited a consistent abundance order across the four wetland types in decreasing order as follows: suaeda wetland > reed wetland > pond returning wetland > tidal flat wetland. Lastly, the relative abundances of Desulfobacterota_I and Desulfobacterota_G_459546 decreased in the following order: tidal flat wetland > suaeda wetland > reed wetland > pond returning wetland.

At the genus level (Figure 3a), the ten most dominant genera were as follows: *RSA9* (3.25% to 4.19%), *SZUA_442* (2.97% to 3.42%), *SP4260* (2.40% to 3.38%), *Gillisia* (2.01% to 3.57%), *Achromobacter* (0.38% to 2.75%), *Nitrosopumilus_5141* (1.16% to 3.48%), *Salinigranum* (0% to 3.64%), *Fodinibius_786578* (0% to 3.34%), *Lysobacter_A_615995* (0.31% to 3.41%), and *JAAXHJ01* (1.65% to 3.20%). *RSA9* was the predominant genus, followed by *SZUA_442* and *SP4260.* The relative abundance of *RSA9* was highest in the tidal flat wetland and significantly exceeded that of the three other wetland types. The relative abundances of *SZUA_442*, *SP4260*, *Achromobacter*, *Nitrosopumilus_5141*, *Salinigranum*, *Fodinibius_786578*, *Lysobacter_A_615995*, and *JAAXHJ01* did not significantly differ between the four wetland types. *Gillisia* and *Lysobacter_A_615995* were significantly more abundant in the suaeda wetland compared with the three other wetland types. The differences in the relative abundance of the top 20 bacterial genera found in all soil samples are visually presented using heatmap analysis (Figure 3a), which reflects the variations in the soil bacterial community structure across the different wetland types. The overall number of bacterial populations in the pond returning and tidal flat wetlands was greater than that in the reed and suaeda wetlands, which suggested that different wetland types influence the distribution of soil microorganisms.

LEfSe analysis was employed to identify the abundance differences in bacterial taxa across the various wetland types. Figure 4 shows that 47 bacterial groups exhibited significant differences, which are specified as follows: the pond returning wetland harbored 17 significantly different bacterial branches, wherein *SIO2C1* and *Marinobacter_A_637054* were characteristic microorganisms at the genus level; the tidal flat wetland contained 7 significantly different branches, wherein *CSSed10_48* was the characteristic microorganism at the genus level; the reed wetland included 13 significantly different branches, wherein *GWC2_73_18* was the characteristic microorganism at the genus level; and the suaeda wetland possessed 10 significantly different branches, wherein *JACTMI01*, *JABFSM01*, *Lysobacter_A_615995*, and *Flavobacterium* were the characteristic microorganisms at the genus level. The LEfSe analysis of the soil from the different wetland types is depicted in Figure 5, and an LDA threshold was set above 4.0. A higher value indicated a more significant contribution of the corresponding taxon to the inter-group differences. The contribution rates of the microbial communities from the different wetland types to the inter-group differences in the various soils were nonuniform. In the pond returning wetland, the contribution rates of the differential bacterial communities to the inter-group differences in descending order at the phylum level were as follows: Proteobacteria > Acidobacteria > Gemmatimonadota > Bacteroidetes. In the tidal flat wetland, the contribution rates at the phylum level in descending order were as follows: Proteobacteria > Deinococcota. In the reed wetland, the contribution rates at the phylum level in descending order were as follows: Chloroflexota > Actinobacteriota > Proteobacteria. In the suaeda wetland, the contribution rates at the phylum level in descending order were as follows: Proteobacteria > Acidobacteriota > Gemmatimonadota > Bacteroidota.

### 3.4. Functional Potential of Soil Bacterial Communities in Different Wetland Types

The functional potential of the soil microbiota across the different wetland types was predicted using PICRUSt based on the KEGG database. The predicted results primarily included six first-level functional categories: metabolism, human disease, organismal systems, environmental information processing, cellular processes, and genetic information processing (Figure 6). Further analysis of the secondary function predictions revealed that the most prevalent second-level functions (KEGG level 2) were signal transduction, biosynthesis of other secondary metabolites, xenobiotics biodegradation and metabolism, metabolism of terpenoids and polyketides, and the endocrine system. Figure 7 illustrates the distribution of the top 20 third-level functions (KEGG level 3) across the different wetland types. The most abundant pathways were valine, leucine, and isoleucine biosynthesis, lipoic acid metabolism, bacterial chemotaxis, synthesis and degradation of ketone bodies, and fatty acid biosynthesis. Among the four wetland types, no significant differences were observed in pathways such as valine, leucine, and isoleucine biosynthesis, lipoic acid metabolism, D-glutamine and D-glutamate metabolism, streptomycin biosynthesis, biosynthesis of amino acids, aminoacyl-tRNA biosynthesis, pantothenate and CoA biosynthesis, protein export, ribosome, citrate cycle (TCA cycle), mismatch repair, and carbon fixation in photosynthetic organisms.

Bacterial chemotaxis, the synthesis and degradation of ketone bodies, fatty acid biosynthesis, the biosynthesis of terpenoids and steroids, biotin metabolism, D-alanine metabolism, and flagellar assembly represented seven functional pathways that exhibited varying distributions across the four wetland types, with an uneven prevalence. Specifically, bacterial chemotaxis was the most abundant in pond returning wetlands, whereas fatty acid biosynthesis, D-alanine metabolism, and flagellar assembly were the least abundant in these same areas. The synthesis and degradation of ketone bodies was the most prevalent in reed wetlands, while biotin metabolism was the most abundant in Suaeda wetlands. These results were derived from predicted functions based on 16S rRNA gene sequences rather than actual metagenomic data, which imposes specific limitations on the functional predictions. Future research will aim to validate these findings using metagenomic data.

## 4. Discussion

### 4.1. Effects of Different Wetland Types on Soil Physicochemical Properties

Different types of wetlands, characterized by varying inputs of litterfall, root exudates, and human disturbances, significantly influence soil physicochemical properties [2,16]. Studies have indicated that these factors are essential for maintaining the ecological health and functions within estuarine wetlands, including carbon sequestration and wetland restoration [29]. Litter is a significant source of soil organic matter in the Liaohe Estuary Wetland. The litter decomposition rates and the nutrient contents of various plant species exhibit considerable variation, which directly influences the nutrient cycling within the soil [30]. Furthermore, agricultural and industrial activities can introduce pollutants, which alter the chemical composition of wetland soils. For instance, applying chemical fertilizers and pesticides may result in elevated concentrations of heavy metals in the soil, thereby seriously threatening the health of wetland ecosystems [31]. In this study, the total carbon and nitrogen contents in the reed and suaeda wetlands were significantly higher than those observed in the pond returning and tidal flat wetlands, which aligns with existing research findings [21,32,33,34]. This disparity may result from the continuous input of organic matter, root exudate deposition, and nutrient release and recycling in the reed and suaeda wetlands. Contrasting with the observed patterns of total carbon and nitrogen, the C/N ratio was significantly lower in the reed wetland than in the suaeda and tidal flat wetlands. Previous research [35,36,37] reported that the nitrogen content was considerably lower in suaeda than in reeds, which may explain the low carbon-to-nitrogen ratio in the reed wetland. A lower soil C/N ratio correlates with a higher nitrogen mineralization rate, which is advantageous for the absorption of nitrogen by microorganisms and plants [38]. Studies have demonstrated that the high carbon-to-nitrogen ratio in reed wetlands is advantageous for soil organic carbon sequestration. Additionally, the soil salinity was significantly higher in tidal flat wetlands than in reed and suaeda wetlands, which aligns with existing research findings [39,40]. However, the soil salinity levels in these three types of wetlands were significantly lower than those observed in pond returning wetlands, likely due to the substantial addition of feed from human aquaculture activities before the abandonment of the ponds for restoration.

### 4.2. Effects of Different Wetland Types on Soil Bacterial Community Diversity

Previous studies have indicated that soil characteristics in different wetlands play a crucial role in altering soil microbial communities [41,42]. Soil pH, salinity, and nutrient contents, particularly the availability of carbon (C), nitrogen (N), potassium (K), and phosphorus (P), are critical factors that significantly affect microbial abundance [15,22,43,44,45,46]. Some studies have proposed a significant negative correlation between soil pH and bacterial diversity [47,48]; however, our study found no significant correlation between soil pH and any bacterial diversity index. This may be attributed to the relatively narrow pH range (8.31–8.49) observed, which hindered our ability to establish any correlation with bacterial diversity. In estuarine wetlands, soil salinity is a crucial indicator that greatly influences the diversity of bacterial communities. Our study identified a significant negative correlation between the Chao1 and Shannon entropy indices and soil salinity, which is consistent with existing research findings [49,50,51,52,53]. Although some studies suggest that bacterial diversity is unaffected by phosphorus levels [54], our study found a highly significant negative correlation between the Shannon entropy index and total phosphorus. High-phosphorus environments may promote the growth of specific functional groups, such as denitrifying bacteria, while inhibiting the richness of others. Additionally, our study revealed a highly significant positive correlation between the observed features index and the total carbon which indicated that carbon is a key driver of bacterial diversity. The presence of carbon in the soil can enhance the reproduction and growth of a broader variety of bacteria, thereby increasing the level of bacterial diversity. This outcome, in turn, enhances the soil fertility and nutrient cycling efficiency and, ultimately, influences the stability and sustainability of the entire ecosystem [7,55,56,57].

### 4.3. Effects of Different Wetland Types on Soil Bacterial Community Composition

Similar to bacterial community diversity, the relative abundances of dominant phyla are influenced by different wetland types [58]. In terms of bacterial composition, distinct wetlands harbor unique bacterial community structures [15]. The most abundant phyla in the Liaohe Estuary wetlands were Proteobacteria (26.43% to 33.92%), Bacteroidota (9.39% to 18.25%), Gemmatimonadota (8.20% to 19.81%), and Actinobacteriota (2.17% to 24.41%), which aligns with prior research findings [59,60,61]. This study revealed that despite the varying wetland types—including “suaeda wetlands”, “reed wetlands”, “pond returning wetlands”, and “tidal flat wetlands”—the compositions of all detected bacterial phyla were similar. Different wetland environments may share a certain degree of similarity, particularly regarding their bacterial community composition. Such similarities may arise from common environmental factors among wetlands, such as water quality, soil characteristics, or vegetation types. Previous studies have indicated that Proteobacteria is the most abundant and widespread phylum found in plant rhizospheres [62], soils [17], and wetlands [41,63] and exhibits strong degradation capabilities. It can decompose organic matter, thereby promoting the breakdown and recycling of organic material in wetlands [64] and maintaining the stability and health of wetland ecosystems. In our study, the relative abundance of Proteobacteria was highest in the suaeda wetlands, which typically exhibit high salinity and alkalinity. These conditions were conducive to the growth of Proteobacteria [65]. Additionally, halophytic herbs provided easily decomposable organic matter and offered a rich nutrient source for microorganisms such as Proteobacteria, which increased its relative abundance.

At the genus level, the bacteria with the highest abundance, such as *RSA9*, *SZUA_442*, *SP4260*, and *Gillisia*, played significant roles in the soil ecosystem of the Liaohe Estuary wetlands by participating in the decomposition of organic matter and nutrient cycling [30,66,67]. *RSA9*, a dominant genus, is commonly found in environments such as soil, the rhizosphere, freshwater, and sediments, where these bacteria may be involved in various biogeochemical cycling processes, including the carbon, nitrogen, and sulfur cycles [68]. Studies have indicated [69,70] that *Gillisia* and *Lysobacter_A_615995* are present in diverse environments, including freshwater, marine sediments, sea ice algae, and seawater. In this study, the abundance of these two bacteria in the suaeda wetland was significantly higher than that in the three other wetland types, which may be attributed to their adaptability to high-salinity environments. Conversely, *SZUA_442*, *SP4260*, *Achromobacter*, *Nitrosopumilus_5141*, *Salinigranum*, *Fodinibius_786578*, *Lysobacter_A_615995*, and *JAAXHJ01* did not significantly differ between the four wetland types, potentially due to their similar adaptability to varying environmental conditions, which may influence the stability and function of the wetland ecosystem. The distinct differentiation and clustering of bacterial communities, as evidenced by the heatmap and NMDS analysis, indicated that different wetland types harbored unique bacterial communities, and suggested that the composition of soil bacterial communities largely depends on the physicochemical characteristics of the soil. This finding aligns with previous research outcomes [71,72]. While our sampling strategy was adequate for preliminary assessments, it may not adequately represent the broader spatial dynamics inherent to each wetland type. We recognize that this limitation could affect the generalizability of our findings. Therefore, we recommend future studies adopt a more extensive and spatially diverse sampling approach to effectively capture the comprehensive variations in microbial communities across different wetland ecosystems.

### 4.4. Impacts of Different Wetland Types on Soil Microbial Community Function

Bacterial functional groups play a crucial role in the Earth’s chemical cycles and energy flux within ecosystems, and their abundance and variation are closely linked to soil nutrient balance and development [73,74]. The PICRUSt analysis revealed that the overall functional characteristics of the bacterial communities across the four wetland types were significantly similar. Fundamental functions, such as amino acid metabolism, carbohydrate metabolism, and membrane transport, were ubiquitous in the soil microbial communities, aligning with previous reports [75,76,77,78]. This study indicated that the relative abundance of valine, leucine, and isoleucine biosynthesis, alongside lipoic acid metabolism, were the top two high-functioning groups that play a vital role in environmental processes. For instance, amino acid biosynthesis is essential to the nitrogen cycle [79], whereas fatty acid synthesis and degradation directly influence the carbon cycle [80], which reflects the high turnover efficiency of the carbon and nitrogen cycles in coastal wetlands. Many bacteria within the Proteobacteria phylum can fix nitrogen and carbon, whereas most bacteria in the Actinobacteriota and Nitrospirae phyla can oxidize nitrite to nitrate for plant assimilation [81,82,83,84]. The *RSA9*, *SZUA_442*, and *SP4260* detected in this study exhibited these functions. Additionally, soil bacteria can influence carbon and nitrogen cycles via other metabolic pathways. Fatty acid biosynthesis and D-alanine metabolism are key metabolic pathways via which soil bacteria contribute to these cycles [85,86,87]. In this study, these two functions exhibited the lowest abundance in the aquaculture pond restoration wetlands, likely due to the ongoing alterations in the microbial community structure during the restoration process from aquaculture ponds. This transition may necessitate considerable time to attain a microbial diversity and functionality comparable to that of natural wetlands. The synthesis and degradation of ketone bodies represent significant metabolic processes in living organisms [88], particularly influencing the carbon metabolism of soil microorganisms. This study discovered that this functional pathway is most abundant in reed wetlands, likely due to the rich organic carbon sources available in these environments, which create favorable conditions for microbial growth. Bacteria such as Chloroflexota, Actinobacteriota, and Proteobacteria can utilize these carbon sources to synthesize ketone bodies. These observations underscore the critical role of wetland bacteria in carbon and nitrogen cycling and emphasize their significance in ecosystem services. The PICRUSt predictions offer valuable insights into the potential functional capabilities of the microbial communities examined in our study. However, we recognize that these predictions are derived from 16S rRNA gene sequences and may not fully encapsulate the functional capacities of the communities. Consequently, we recommend future studies utilize metagenomic approaches to validate these findings and achieve a more comprehensive understanding of the microbial functions.

## 5. Conclusions

This study utilized high-throughput sequencing and PICRUSt analysis to investigate the soil nutrients, bacterial structure, and potential metabolic functions of four distinct wetland types in the Liaohe Estuary Wetland. The physicochemical properties of the soils in the various wetland types significantly influenced the structure and function of the bacterial communities. The key factors included water content, soil electrical conductivity, salinity, total phosphorus, and total carbon. While the soil bacterial composition across the four wetland types was similar at the phylum level, it varied in abundance. Proteobacteria and Bacteroidota were identified as the dominant phyla, while *RSA9* and *SZUA_442* emerged as the predominant genera at the genus level. The PICRUSt predictive analysis revealed the functional potential of the bacterial communities, highlighting valine, leucine, and isoleucine biosynthesis, lipoic acid metabolism, bacterial chemotaxis, the synthesis and degradation of ketone bodies, and fatty acid biosynthesis as the most abundant pathways. These functional pathways are crucial for carbon and nitrogen cycling. This study enhances our understanding of soil microbial communities within estuarine wetland ecosystems.

## Figures and Tables

**Figure 1 microorganisms-12-02075-f001:**
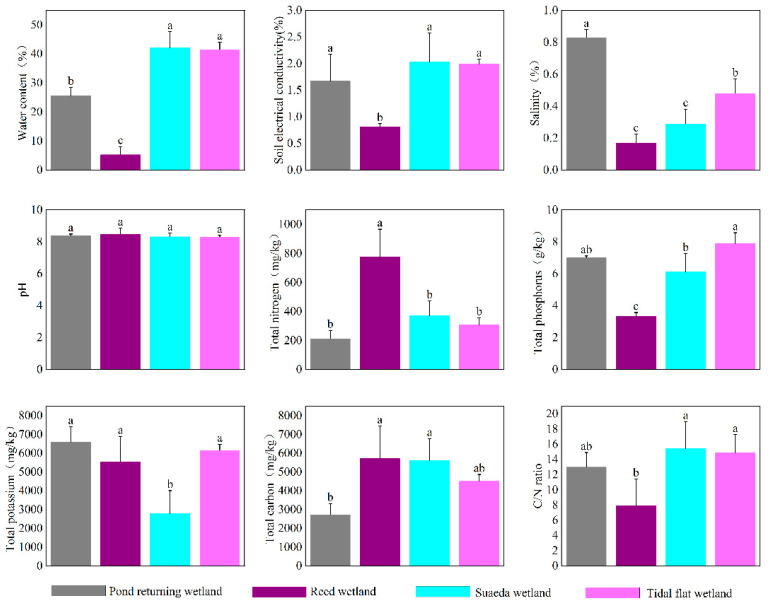
Soil physicochemical properties in different wetland types. Data are presented as mean values ± standard error (*n* = 3). Different letters (a, b, c) within the same row indicate significant differences; *p* < 0.05.

**Figure 2 microorganisms-12-02075-f002:**
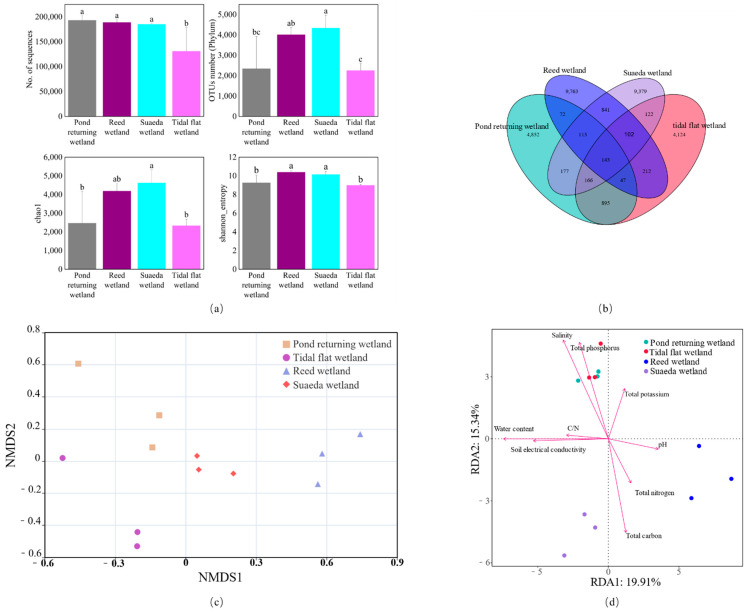
The effective sequence information detected in the various wetland types, the number of OTUs at the phylum level, Chao1 index, and Shannon index (**a**), with data shown as mean values ± standard error (n = 3). Different letters (a, b, c) within the same row indicate significant differences at *p* < 0.05. A Venn diagram of soil bacterial OTUs is depicted in (**b**). The non-metric multidimensional scaling analysis (NMDS) is illustrated in (**c**). The redundancy analysis (RDA) of the soil bacterial communities with respect to the soil physicochemical properties is shown in (**d**).

**Figure 3 microorganisms-12-02075-f003:**
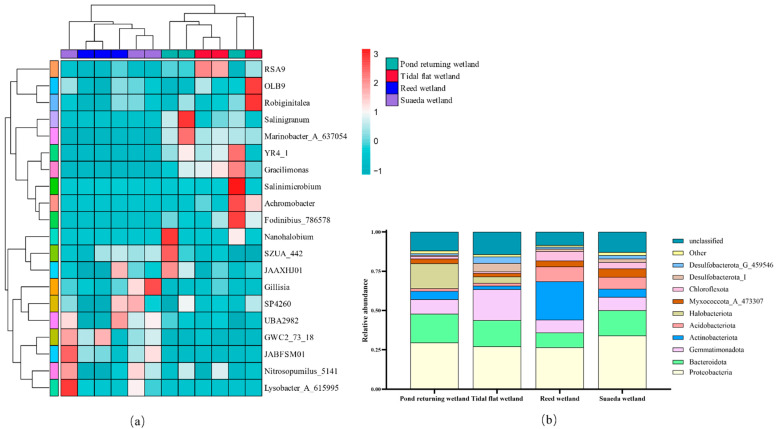
Abundance of dominant bacterial genera (**a**) and phyla (**b**) in the soil of different wetland types.

**Figure 4 microorganisms-12-02075-f004:**
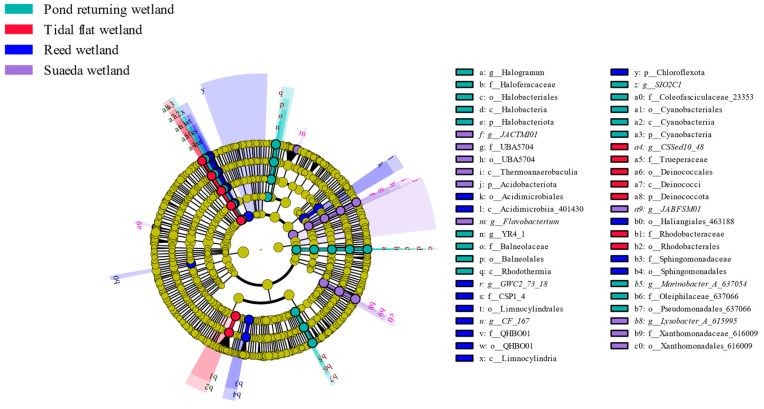
LEfSe analysis reveals significant variations in bacterial taxa between different wetland types. It presents a cladogram illustrating these differences in soil bacterial abundance across the various wetland types. In the evolutionary branch diagram, the circular radiation emanating from the center represents the taxonomic ranks of phyla and genera, and each small circle indicates the position of a taxonomic rank within different classification levels, with the circle diameter proportional to the relative abundance. Species exhibiting no significant differences are uniformly colored yellow, while others are colored according to their highest abundance.

**Figure 5 microorganisms-12-02075-f005:**
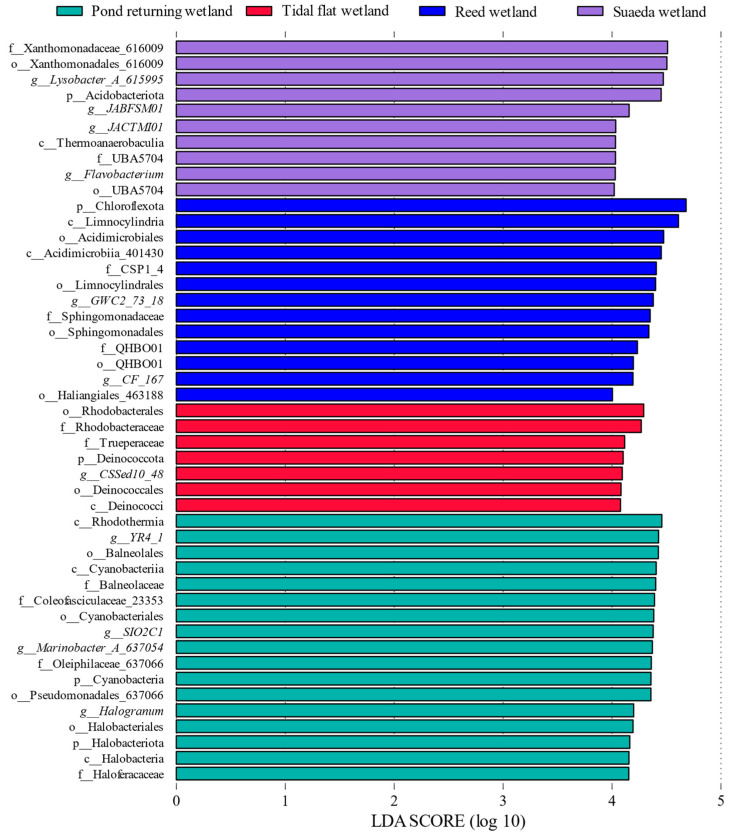
The LDA effect size bar plots illustrate differentially abundant bacterial taxa in the wetlands of the Liaohe Estuary. These plots present the LEfSe analysis LDA bar chart, where the vertical axis denotes the taxonomic units that are significantly different between groups, while the horizontal axis visualizes the logarithmic scores from the LDA difference analysis corresponding to each classification group. The data are sorted by score magnitude to convey the differences observed. The length of each bar represents the extent of the difference between the grouped samples; thus, a longer bar indicates a more significant difference between the taxonomic units. Additionally, the varying colors in the bar chart correspond to the sample groups in which each taxonomic unit is more abundant.

**Figure 6 microorganisms-12-02075-f006:**
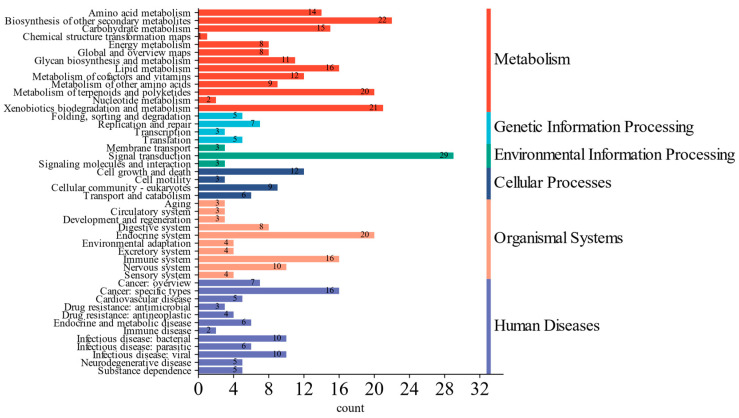
Primary and secondary metabolic functions of soil bacterial communities.

**Figure 7 microorganisms-12-02075-f007:**
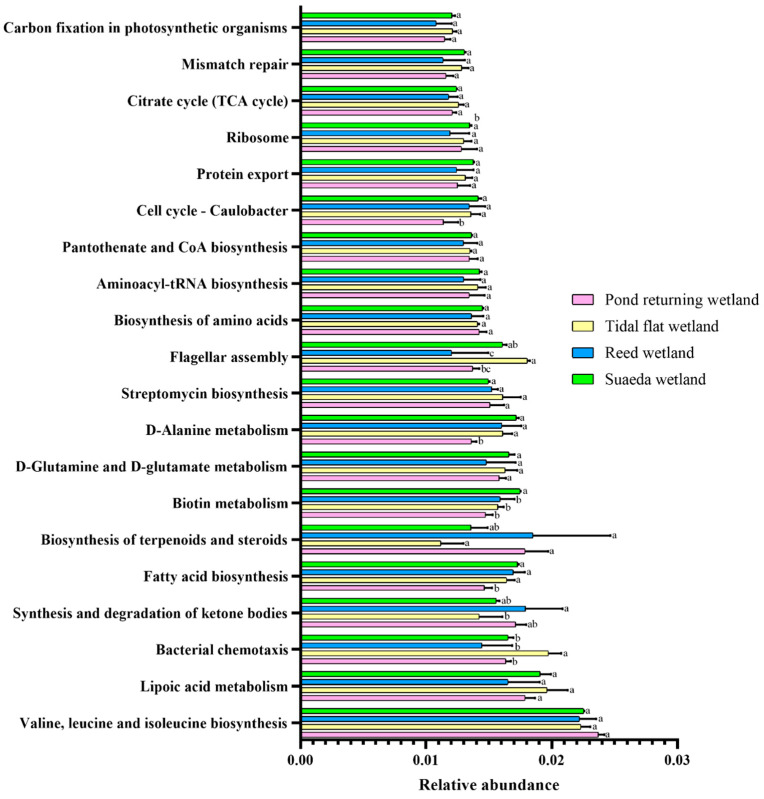
Tertiary metabolic functions of soil bacterial communities. The bar graph illustrates the relative abundances of major metabolic functions within soil bacterial communities across various wetland types. Different letters (a, b) in the same row signify significant differences at *p* < 0.05.

## Data Availability

The data underlying this research are available in this article.

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
