# Peer review of "Structure and Function of Soil Bacterial Communities in the Different Wetland Types of the Liaohe Estuary Wetland"

_microorganisms, 2024, doi:10.3390/microorganisms12102075_

Round 1
Reviewer 1 Report
Comments and Suggestions for Authors
The present work contributes to understand the diversity and microbial composition in coastal zone in terms of their possible role played in the ecosystem dynamic. In my opinion the authors have made a comprehensive discussion of their results, and in order to improve the manuscript, these are my suggestions:
i) Line 24 – please italicize Pseudomonas
ii) Line 24 – regarding the word “Bacilli”, what is the meaning? Genera Bacillus (italics) or rods? I interpreted as rods, so it must be written lowercase.
iii) Line 68 – please include comma between [16] and pH
iv) Line 101 – sampling was performed in September. It makes the diversity study as punctual in place of seasonal. Please inform weather conditions in September 2023 (average temperature, pluviosity, duration of days and humidity). Additionally, it is important to mention that the authors did not assess community variation among seasons. I suggest including a short paragraph in discussion by saying this information as well as to compare to diversity studies during late summer in temperate zones.
v) Lines 112-120 – it is important to include references for methods for the soil characterization: water content; pH and salinity; TC; TN, TOC; TP; and TK
vi) Line 114 – please correct Ph to pH
vii) Lines 156 and 193 – the letter p in lowercase
viii) All the paragraph started in line 223 – it is mandatory to italicize all genera
ix) Figure 4, please italicize all genera in the second column
x) Figure 5, the same as in figure 4 regarding genera
xi) Line 314 – Studies in place of tudies
xii) Line 316 – please include at least 2 more references. In line 314 readers are informed about studies that have indicated external factors for estuarine zones manutention
xiii) Line 337 – why haven’t you mentioned potassium? It needs to be mentioned.
xiv) Line 363 – please include references regarding the “…prior research findings”
xv)Line 380 – italicize Gillisia
xvi) Lines 385-390 – same as suggestion viii (all genera must be italicized)
xvii) Line 436 – Proteobacteria in place of proteobacteria
xviii) Line 437 – Bacteroidota in place of bacteroidota
Author Response
- i) Line 24 – please italicize Pseudomonas.
Dear Reviewer, thank you very much for pointing this out. We have corrected the error in the revised manuscript, where you can see in lines 23-24 that "The dominant bacterial phylum identified is Proteobacteria and Bacteroidota." Pseudomonas should not appear here; it was our mistake, and we apologize.
- ii) Line 24 – regarding the word “Bacilli”, what is the meaning? Genera Bacillus (italics) or rods? I interpreted as rods, so it must be written lowercase.
Dear Reviewer, thank you very much for pointing this out. We have corrected the error in the revised manuscript, where you can see in lines 23-24 that "The dominant bacterial phylum identified is Proteobacteria and Bacteroidota." Bacilli should not appear here; it was our mistake, and we apologize.
iii) Line 68 – please include comma between [16] and pH.
Dear Reviewer, thank you very much for pointing this out. We have made the correction in the revised manuscript, as can be seen on line 68.
- iv) Line 101 – sampling was performed in September. It makes the diversity study as punctual in place of seasonal. Please inform weather conditions in September 2023 (average temperature, pluviosity, duration of days and humidity). Additionally, it is important to mention that the authors did not assess community variation among seasons. I suggest including a short paragraph in discussion by saying this information as well as to compare to diversity studies during late summer in temperate zones.
Dear Reviewer, thank you very much for pointing this out. We have made the necessary revisions in the revised manuscript, specifically in lines 108-101, where we have added the relevant weather conditions. We agree with your suggestion about the importance of seasonal variation in bacterial communities, and this will be an important area of research for us in the future.
- v) Lines 112-120 – it is important to include references for methods for the soil characterization: water content; pH and salinity; TC; TN, TOC; TP; and TK.
Dear Reviewer, thank you very much for pointing this out. We have made the necessary revisions in the revised manuscript, specifically in lines 126-130, where we have added the corresponding references.
- vi) Line 114 – please correct Ph to pH.
Dear Reviewer, thank you very much for pointing this out. We have made the correction in the revised manuscript, as can be seen on line 124.
vii) Lines 156 and 193 – the letter p in lowercase.
Dear Reviewer, thank you very much for pointing this out. We have made the corrections in the revised manuscript, as can be seen on lines 167 and 205.
viii) All the paragraph started in line 223 – it is mandatory to italicize all genera.
Dear Reviewer, thank you very much for pointing this out. We have made the necessary revisions in the revised manuscript, specifically from line 223 onwards.
- ix) Figure 4, please italicize all genera in the second column.
Dear Reviewer, thank you very much for pointing this out. We have made the necessary revisions, and you can see the italicized genera in the second column of Figure 4 on page 7 of the revised manuscript.
- x) Figure 5, the same as in figure 4 regarding genera.
Dear Reviewer, thank you very much for pointing this out. We have made the necessary revisions, and you can see the italicized genera in Figure 5 on page 8 of the revised manuscript.
- xi) Line 314 – Studies in place of tudies.
Dear Reviewer, thank you very much for pointing this out. We have made the correction in the revised manuscript, as can be seen on line 335.
xii) Line 316 – please include at least 2 more references. In line 314 readers are informed about studies that have indicated external factors for estuarine zones manutention.
Dear Reviewer, thank you very much for pointing this out. We have added the necessary references in the revised manuscript, specifically in lines 338-343.
xiii) Line 337 – why haven’t you mentioned potassium? It needs to be mentioned.
Dear Reviewer, thank you very much for pointing this out. We have included potassium in the revised manuscript, as can be seen on line 365, and have also improved the references.
xiv) Line 363 – please include references regarding the “…prior research findings”.
Dear Reviewer, thank you very much for pointing this out. We have added the necessary references in the revised manuscript, specifically on line 391 (references 59-61).
- xv) Line 380 – italicize Gillisia.
Dear Reviewer, thank you very much for pointing this out. We have italicized Gillisia in the revised manuscript, as can be seen on line 391.
xvi) Lines 385-390 – same as suggestion viii (all genera must be italicized).
Dear Reviewer, thank you very much for pointing this out. We have italicized all genera in the revised manuscript, as can be seen from lines 410 to 418.
xvii) Line 436 – Proteobacteria in place of proteobacteria.
Dear Reviewer, thank you very much for pointing this out. We have corrected Proteobacteria in the revised manuscript, as can be seen on line 472.
xviii) Line 437 – Bacteroidota in place of bacteroidota.
Dear Reviewer, thank you very much for pointing this out. We have corrected Bacteroidota in the revised manuscript, as can be seen on line 473.

Reviewer 2 Report
Comments and Suggestions for Authors
The manuscript deals with an important issue, because the interactions between microorganisms in different environments is the key to understanding many aspects troubling scientists these days. The paper, however, needs to be revised, my comments below:
- No appendix available despite references to it in the paper
- Line 51-54: the authors state that in recent years a lot of research has been on the microbiome of the vetlands and the interactions between them, so I ask where is the novelty in this work
- Line 127: spelling error
- Figure 2, 4, 5 please rewrite figure titles
- Line 223-224: there is no such bacteria type as RSA9 and SZUA_442, SP4260, please correct or clarify it
- Line 254-260: the mentioned systematic units are not genus, please correct it
- Figure 4 and 5 what is the point of giving to graphics readings of systematic units that tell the reader nothing, such as g_JACTM101? Please explain or change them
- Figure 7: please add to the description a and b legend
- Line 314: spelling error
- Line 340: spelling error
- Line 363: please add quotes
- Line 370: please use proper systematic name
- Line 379-397: please correct generic names, as in earlier stages, also use correct systematic spelling, including italics
- Line 405-408: the most important metabolic properties described are not adequately explained, what is their role on the environment, how they are determined at the molecular level in microorganisms, please add literature citations
- Line 411-413: if these are the main functional pathways in microoroganisms, please explain them and not just list them, moreover literature citations are missing
- Line 417-423 this information is given above, please combine these fragments so as not to duplicate information
- Line 428-430: please rephrase this sentence, because these are not “Bacteria” but a specific systematic unicellular, in addition, please add literature citations
- Why sector 4.4 does not describe other results obtained from the analysis of PICRUSt, which were identified in significant numbers, as shown in figure 6
- Line 432-433: why the authors write that the bacteria show significant differences, while the results presented and described in the paper show that there are no significant differences between the studied activities, which the authors themselves emphasize, please rephrase and expand this sentence
- Line 436-437: please use the correct spelling of the systematic names, please once again explain what the generic names mean
- Line 437-441: briefly describe the metabolic reactions mentioned, the mere mention in the subheading does not tell the reader anything, it is not an inference but a repetition of the results.
Author Response
1- No appendix available despite references to it in the paper
Dear Reviewer,
Thank you for pointing out this issue. We have noticed that there is no appendix in the paper. The misunderstanding arises from a mistake in our word choice during the writing of the paper, where we incorrectly referred to figures as appendices. This may have conveyed the wrong information to you, leading you to believe that there is an appendix. We sincerely apologize for any inconvenience this may have caused during your review. We will be more careful with our word choice in future papers to avoid such mistakes.
2- Line 51-54: the authors state that in recent years a lot of research has been on the microbiome of the vetlands and the interactions between them, so I ask where is the novelty in this work
Dear Reviewer,
We appreciate your inquiries regarding the novelty of this study. We believe that the uniqueness of our research is primarily evident in the following aspects:
Comprehensive comparative study: We conducted an extensive comparison of soil bacterial communities across four distinct wetland types within the same region, specifically the Liaohe Estuary Wetland. This comparative analysis of environmental gradients is relatively novel in the context of existing literature.
In-depth analysis of the impact of environmental gradients: Our study elucidates how environmental gradients, ranging from tidal plains to reed wetlands, suaeda wetlands, and ponds, influence the structure and function of microbial communities. This understanding is crucial for comprehending the biogeochemical cycles within wetland ecosystems.
Ecological and environmental significance: The findings of our research offer a scientific foundation for wetland protection, restoration, and the formulation of environmental policies, particularly against the backdrop of global climate change and biodiversity conservation. These results hold substantial practical implications.
3- Line 127: spelling error
Dear Reviewer,
Thank you very much for pointing this out. We have noticed this issue and have made the necessary corrections, which can be seen on line 137 of the revised manuscript.
4- Figure 2, 4, 5 please rewrite figure titles
Dear Reviewer,
Thank you very much for pointing this out. We have noticed this issue and have made the necessary corrections, which can be seen on pages 5, 7, and 8 of the revised manuscript.
5- Line 223-224: there is no such bacteria type as RSA9 and SZUA_442, SP4260, please correct or clarify it
Dear Reviewer,
Thank you for your observation regarding the OTUs RSA9, SZUA_442, and SP4260 mentioned in our manuscript. Following your feedback, we have conducted further verification and clarification of their taxonomic positions. Here is the updated information:
RSA9 is classified as:
Domain: Bacteria (k__Bacteria)
Phylum: Gemmatimonadota (p__Gemmatimonadota)
Class: Gemmatimonadetes (c__Gemmatimonadetes)
Order: Longimicrobiales (o__Longimicrobiales)
Family: RSA9 (f__RSA9)
Genus: RSA9 (g__RSA9)
Species: RSA9_sp003242735 (s__RSA9_sp003242735)
SZUA_442 is classified as:
Domain: Bacteria (k__Bacteria)
Phylum: Actinobacteriota (p__Actinobacteriota)
Class: Acidimicrobiia_402965 (c__Acidimicrobiia_402965)
Order: UBA5794 (o__UBA5794)
Family: UBA5794 (f__UBA5794)
Genus: SZUA_442 (g__SZUA_442)
Species: SZUA_442_sp003235475 (s__SZUA_442_sp003235475)
SP4260 is classified as:
Domain: Bacteria (k__Bacteria)
Phylum: Proteobacteria (p__Proteobacteria)
Class: Gammaproteobacteria (c__Gammaproteobacteria)
Order: Woeseiales (o__Woeseiales)
Family: Woeseiaceae (f__Woeseiaceae)
Genus: SP4260 (g__SP4260)
Species: Not designated (s__Not designated)
We appreciate your valuable comments which have contributed to enhancing the accuracy of our research findings.
6- Line 254-260: the mentioned systematic units are not genus, please correct it
Dear Reviewer,
Thank you for your observation. We have noted the issue with the classification of the systematic units mentioned. The corrections have been made and can be reviewed on lines 270-274 of the revised manuscript.
7- Figure 4 and 5 what is the point of giving to graphics readings of systematic units that tell the reader nothing, such as g_JACTM101? Please explain or change them
Dear Reviewer,
Thank you for your insightful comment. We have revised the figure titles and descriptions for clarity.
LEfSe analysis was conducted to identify differences in the abundance of bacterial taxa across various wetland types. Figure 4 illustrates that 47 bacterial groups showed significant differences. Specifically:
The pond returning wetland had 17 significantly different bacterial branches, among which g__SIO2C1 and g__Marinobacter_A_637054 are characteristic microbes at the genus level.
The tidal flat wetland contained 7 significantly different branches, with g__CSSed10_48 being a characteristic microbe at the genus level.
The reed wetland included 13 significantly different branches, with g__GWC2_73_18 identified as a characteristic microbe at the genus level.
The suaeda wetland had 10 significantly different branches, featuring characteristic microbes at the genus level such as g__JACTMI01, g__JABFSM01, g__Lysobacter_A_615995, and g__Flavobacterium.
8- Figure 7: please add to the description a and b legend
Dear Reviewer,
Thank you for your observation. We have added the necessary legends for sections a and b of Figure 7 to enhance clarity. The revisions can be found on line 328 of the revised manuscript.
9- Line 314: spelling error
Dear Reviewer,
We appreciate your attention to detail. The spelling error on line 314 has been corrected. The correction can be reviewed on line 334 of the revised manuscript.
10- Line 340: spelling error
Dear Reviewer,
Thank you for pointing out the spelling error. It has been corrected and the updated text can be found on line 367 of the revised manuscript.
11- Line 363: please add quotes
Dear Reviewer,
We have incorporated your suggestion to add quotes for clarity. The change has been made and is reflected on lines 391-392 of the revised manuscript.
12- Line 370: please use proper systematic name
Dear Reviewer,
Thank you for your guidance on the use of systematic names. We have updated the text with the correct systematic nomenclature, which can now be seen on line 397 of the revised manuscript.
13- Line 379-397: please correct generic names, as in earlier stages, also use correct systematic spelling, including italics
Dear Reviewer,
Thank you for pointing out this issue. We have corrected the generic names and ensured the use of proper systematic spelling, including italics, as per the earlier stages. The changes can be seen on lines 406-417 of the revised manuscript..
14- Line 405-408: the most important metabolic properties described are not adequately explained, what is their role on the environment, how they are determined at the molecular level in microorganisms, please add literature citations
Dear Reviewer,
Thank you very much for pointing out this issue. We have taken note of the problem and have made the necessary revisions. You can see the changes in lines 432-438 of the revised manuscript.
15- Line 411-413: if these are the main functional pathways in microoroganisms, please explain them and not just list them, moreover literature citations are missing
Dear Reviewer,
Thank you very much for your valuable feedback. We have taken note of this issue and have made the necessary revisions. The changes can be found in lines 442-450 of the revised manuscript.
16- Line 417-423 this information is given above, please combine these fragments so as not to duplicate information
Dear Reviewer,
Thank you very much for your valuable feedback. We have taken note of this issue and have made the necessary revisions. The changes can be found in lines 442-450 of the revised manuscript.
17 - Line 428-430: please rephrase this sentence, because these are not “Bacteria” but a specific systematic unicellular, in addition, please add literature citations
Dear Reviewer,
Thank you very much for pointing out this issue. We have taken note of it and have made the necessary revisions. The changes can be found in lines 457-460 of the revised manuscript.
18- Why sector 4.4 does not describe other results obtained from the analysis of PICRUSt, which were identified in significant numbers, as shown in figure 6
Dear Reviewer,
Thank you very much for your insightful comments. We have taken note of the issue you raised. Our focus was on discussing functions that are most directly related to the research hypotheses and objectives. These hypotheses and objectives center on understanding the impact of specific wetland types on the functional potential of soil bacterial communities, particularly processes closely related to carbon and nitrogen cycles. Therefore, we have selected some key metabolic pathways that are directly related to these cycles for detailed discussion, such as the biosynthesis of valine, leucine, and isoleucine, as well as fatty acid metabolism. Although the PICRUSt analysis predicted many other functions, not all predicted functions are directly related to our research objectives. Your raised issue is very beneficial to us, and we will discuss it within our research group to consider our next steps in the research plan.
19- Line 432-433: why the authors write that the bacteria show significant differences, while the results presented and described in the paper show that there are no significant differences between the studied activities, which the authors themselves emphasize, please rephrase and expand this sentence
Dear Reviewer,
Thank you very much for bringing this to our attention. We have recognized the error in our initial expression and have made the necessary corrections. The revisions can be found in lines 469-470 of the revised manuscript.
20- Line 436-437: please use the correct spelling of the systematic names, please once again explain what the generic names mean
Dear Reviewer,
Thank you very much for your valuable feedback. We have taken note of the issue you pointed out and have made the appropriate revisions. The changes are now reflected in lines 471-472 of the revised manuscript.
21- Line 437-441: briefly describe the metabolic reactions mentioned, the mere mention in the subheading does not tell the reader anything, it is not an inference but a repetition of the results.
Dear Reviewer,
Thank you very much for your valuable feedback. We have taken note of the issue you raised, and as a result, we have made the necessary revisions. These changes can now be found in lines 472-476 of the revised manuscript.

Reviewer 3 Report
Comments and Suggestions for Authors
The article is devoted to studying microbial biodiversity in estuaries, using the Liaohe Estuary Wetland model. Four different habitats were sampled and metagenomic sequencing was performed. The species composition of the communities and the relationship between the microbial richness and the amount of carbon in the soil were revealed.
Many results are presented in the paper, but they are not structured to make sense of the many examined metagenomes.
1. I do not find a hypothesis to follow and prove in the article. The authors say in the introduction that the purpose is to elucidate the structure, diversity, and function of soil bacterial communities across four typical wetland types in the Liaohe Estuary Wetland. This is a topic to which over 1090 articles have been devoted (according to Scholar Google), and it does not seem new, on the contrary, the results fully align with prior research findings. The research lacks novelty and significance.
2. The authors refer to genera 'RSA9', 'SZUA_442', and 'SP4260'. Please indicate which genera of microorganisms these are.
3. Appendices are missing from the article. The mentioned "appendix" is missing, as well as supplementary. Thus the data cannot be examined in full.
4. The sequences (both analyzed and raw) are not deposited in a database. It is good that they have an accession number before publication so that there is an opportunity to check the data.
5. The authors analyzed 12 samples. On what basis are the metagenomic data then pooled, and is there a difference between samples of the same type themselves?
6. The discussion should include a clearer connection between the detected 'metabolic functions' and the species or genera of bacteria found in the soil samples. At the phylum level, it is too broad a generalization.
7. Where do you see the place of soil rhizobacteria? They are an interesting class of microorganisms that are of great importance to ecosystems, and you have not mentioned them at all.
8. Conclusions should be revised. The sentence "The dominant phyla identified are proteobacteria and bacteroidota, with the genera RSA9 and SZUA_442 being particularly predominant" does not carry any comprehensive information about the microbial biodiversity of the estuaries.
Comments on the Quality of English LanguageEnglish needs extensive editing.
Author Response
Dear Esteemed Reviewer,
Thank you for your constructive suggestions. In response to your feedback, we have polished the English writing of our manuscript to ensure clarity and precision.
- I do not find a hypothesis to follow and prove in the article. The authors say in the introduction that the purpose is to elucidate the structure, diversity, and function of soil bacterial communities across four typical wetland types in the Liaohe Estuary Wetland. This is a topic to which over 1090 articles have been devoted (according to Scholar Google), and it does not seem new, on the contrary, the results fully align with prior research findings. The research lacks novelty and significance.
Dear Esteemed Reviewer,
Thank you very much for your valuable feedback. We have taken note of the issue you raised, and as a result, we have made the necessary revisions. We have articulated our hypotheses as follows: "We aimed to test the hypotheses that there are significant variations in soil bacterial community structure and function among different wetland types, and that these differences are closely associated with specific environmental factors." These revisions can be found in lines 90-94 of the revised manuscript.
And thank you for raising the question regarding the novelty of our study. We believe that the novelty of our research is primarily reflected in the following aspects:
Comprehensive comparative study: We conducted an integrated comparison of soil bacterial communities across four different wetland types within the same region (the Liaohe Estuary Wetland). This kind of multi-wetland type and multi-environmental gradient comparative study is relatively novel in existing research.
In-depth analysis of environmental gradient impacts: Our study reveals how the environmental gradient from tidal flats to reed wetlands, and then to salt marshes and pond restoration wetlands, affects the structure and function of microbial communities. This is significant for understanding the biogeochemical cycles in wetland ecosystems.
Ecological and environmental significance: The results of our study provide a scientific basis for wetland conservation, restoration, and environmental policy-making. They are particularly relevant in the context of global climate change and biodiversity conservation, offering important practical implications.
We have introduced these points in lines 83-89 of the manuscript.
- The authors refer to genera 'RSA9', 'SZUA_442', and 'SP4260'. Please indicate which genera of microorganisms these are.
Dear Reviewer,
Thank you for your observation regarding the OTUs RSA9, SZUA_442, and SP4260 mentioned in our manuscript. Following your feedback, we have conducted further verification and clarification of their taxonomic positions. Here is the updated information:
RSA9 is classified as:
Domain: Bacteria (k__Bacteria)
Phylum: Gemmatimonadota (p__Gemmatimonadota)
Class: Gemmatimonadetes (c__Gemmatimonadetes)
Order: Longimicrobiales (o__Longimicrobiales)
Family: RSA9 (f__RSA9)
Genus: RSA9 (g__RSA9)
Species: RSA9_sp003242735 (s__RSA9_sp003242735)
SZUA_442 is classified as:
Domain: Bacteria (k__Bacteria)
Phylum: Actinobacteriota (p__Actinobacteriota)
Class: Acidimicrobiia_402965 (c__Acidimicrobiia_402965)
Order: UBA5794 (o__UBA5794)
Family: UBA5794 (f__UBA5794)
Genus: SZUA_442 (g__SZUA_442)
Species: SZUA_442_sp003235475 (s__SZUA_442_sp003235475)
SP4260 is classified as:
Domain: Bacteria (k__Bacteria)
Phylum: Proteobacteria (p__Proteobacteria)
Class: Gammaproteobacteria (c__Gammaproteobacteria)
Order: Woeseiales (o__Woeseiales)
Family: Woeseiaceae (f__Woeseiaceae)
Genus: SP4260 (g__SP4260)
Species: Not designated (s__Not designated)
We appreciate your valuable comments which have contributed to enhancing the accuracy of our research findings.
- Appendices are missing from the article. The mentioned "appendix" is missing, as well as supplementary. Thus the data cannot be examined in full.
Dear Esteemed Reviewer,
Thank you very much for bringing this to our attention. We have noted the issue you pointed out. There are no attachments to this manuscript. The confusion arose due to an error in our word choice during manuscript preparation. In some instances, we mistakenly referred to figures as "appendices," which may have conveyed the wrong information and led you to believe that there were attachments included. We sincerely apologize for any inconvenience this may have caused during your review process. We assure you that we will be more cautious with our terminology in future manuscript submissions to avoid such errors.
- The sequences (both analyzed and raw) are not deposited in a database. It is good that they have an accession number before publication so that there is an opportunity to check the data.
Dear Esteemed Reviewer,
Thank you very much for your observation. We have taken note of this issue and will ensure that the sequencing data is uploaded to the appropriate database at a later stage.
- The authors analyzed 12 samples. On what basis are the metagenomic data then pooled, and is there a difference between samples of the same type themselves?
Dear Esteemed Reviewer,
Thank you very much for your observation. We have taken note of the issue you raised. In our study, we indeed analyzed 12 soil samples, which encompass four different wetland types of the Liaohe Estuary Wetland. Each wetland type is represented by three samples that are spatially adjacent to ensure as much environmental similarity as possible. The objective of our research is to explore the impact of different types of wetlands on the structure and function of soil bacterial communities.
- The discussion should include a clearer connection between the detected 'metabolic functions' and the species or genera of bacteria found in the soil samples. At the phylum level, it is too broad a generalization.
Dear Esteemed Reviewer,
Thank you very much for your valuable feedback. We have taken note of the issue you raised. In our study, the strains RSA9, SZUA_442, and SP4260 were indeed found to possess these functions. We have added supplementary descriptions in the revised manuscript, which can be found in lines 440-441.
- Where do you see the place of soil rhizobacteria? They are an interesting class of microorganisms that are of great importance to ecosystems, and you have not mentioned them at all.
Dear Esteemed Reviewer,
Thank you very much for your insightful comments. We have taken note of the issue you raised. Given that our study sites include tidal flat wetland and pond returning wetland which have fewer or no rhizosphere microbes, this aspect was not covered in our paper. We agree that rhizosphere bacteria represent a valuable area for further research within wetland ecosystems. In future studies, we will consider directly investigating rhizosphere bacterial communities and how they respond to different environmental conditions and plant interactions. We appreciate your valuable guidance.
- Conclusions should be revised. The sentence "The dominant phyla identified are proteobacteria and bacteroidota, with the genera RSA9 and SZUA_442 being particularly predominant" does not carry any comprehensive information about the microbial biodiversity of the estuaries.
Dear Esteemed Reviewer,
Thank you very much for your valuable feedback. We have taken note of the issue you raised and have revised our conclusions accordingly. The modifications can be found in lines 469-472 of the revised manuscript.

Round 2
Reviewer 1 Report
Comments and Suggestions for Authors
Dear authors,
The manuscript has been improved and all suggested modifications were accepted.
Author Response
Comments 1:Dear authors,
The manuscript has been improved and all suggested modifications were accepted.
Response 1:
Dear reviewer,
We would like to express our sincere gratitude to you and the reviewers for your valuable feedback and guidance throughout the review process. The constructive comments have significantly contributed to enhancing the quality and clarity of our research.
We have carefully reviewed and implemented all the recommended changes, and we believe that the final version of our manuscript now more effectively communicates our findings and conclusions.
Best regards.

Reviewer 3 Report
Comments and Suggestions for Authors
The authors have responded to all comments. The manuscript has been greatly improved and I consider it ready for publication.
Comments on the Quality of English LanguageMinor editing of English language required.
Author Response
Comments 1:The authors have responded to all comments. The manuscript has been greatly improved and I consider it ready for publication.
Dear reviewer,
We would like to express our sincere gratitude to you and the reviewers for your valuable feedback and guidance throughout the review process. The constructive comments have significantly contributed to enhancing the quality and clarity of our research.
We have carefully reviewed and implemented all the recommended changes, and we believe that the final version of our manuscript now more effectively communicates our findings and conclusions.
Best regards.
Comments 2:Minor editing of English language required.
Dear reviewer,
Thank you for your feedback regarding the minor editing of the English language required for our manuscript titled "The Structure and Function of Soil Bacterial Communities in Different Wetland Types of the Liaohe Estuary Wetland."
We have taken your comments into consideration and have carefully revised the manuscript to ensure that the language is clear, concise, and academically sound. A professional English language editor has also reviewed the entire document to correct any linguistic inconsistencies and to enhance the overall readability.
The revised manuscript has been thoroughly checked for grammatical errors, typos, and awkward phrasings. We believe the current version has significantly improved in terms of language and presentation.
Thank you once again for your guidance and support.
Best regards.
